


# Return levels of extreme European windstorms, their dependency on the NAO, and potential future risks

Matthew D. K. Priestley[1], David B. Stephenson[1], Adam A. Scaife[2,1], Daniel Bannister[3], Christopher J. T. Allen[4], and David Wilkie[4]

[1]Department of Mathematics and Statistics, University of Exeter, UK
[2]Met Office, Exeter, UK
[3]WTW Research Network, WTW, UK
[4]Model Research & Evaluation, Gallagher Re, UK

**Correspondence:** Matthew D. K. Priestley (m.priestley@exeter.ac.uk)

**Abstract.** Windstorms are the most damaging natural hazard across western Europe. Risk modellers are limited by the observational data record to only ∼60 years of comprehensive reanalysis data that is dominated by considerable inter-annual variability. This makes estimating return periods of rare events difficult and sensitive to choice of historical period used. This study proposes a novel statistical method for estimating wind gusts across Europe based on observed windstorm footprints from the WISC project. Estimates of the 10-year and 200-year return levels are provided. The North Atlantic Oscillation (NAO) is particularly important for modulating lower return levels and setting the tail location parameter, with a less detectable influence on rarer extremes and the tail scale parameter. The optimal length of historical data required to make an accurate return level estimation is quantified using both observed and simulated timeseries of the historical NAO. For estimating 200-year return levels, a data catalogue of at least 20 years is required. For lower return levels the NAO has a stronger influence on estimated return levels and so there is more variability in estimates. Using recent estimates of plausible future NAO states, return levels are largely outside the historical uncertainty, indicating significant increases in risk potential from windstorms in the next 100 years. Our method presents a framework for assessing high return period losses across a range of hazards without the additional complexities of a full catastrophe model.

## 1 Introduction

Extratropical cyclones (ETCs) are the dominant weather system across Europe in the winter season, contributing most of the precipitation (Hawcroft et al., 2012) and strong winds (Ulbrich et al., 2001). ETCs are commonly associated with extreme wind speeds (also known as windstorms) that can impact infrastructure, agriculture, transport, and cause loss of life (e.g. Browning, 2004; Schwierz et al., 2010; Kendon and McCarthy, 2015). The strongest storms such as Lothar (26/12/1999) and Kyrill (18/1/2007) caused total insured losses exceeding €5.5 billion (2022 equivalent; PERILS AG, 2022), with economic losses far exceeding this.

Although long-range predictability of European windstorms exists (e.g. Scaife et al., 2014; Befort et al., 2019; Degenhardt



et al., 2022), extreme events are simulated based on historical events and known natural variability. The main tools used to estimate rare events and their risk potential are catastrophe models (Grossi and Kunreuther, 2005). These models generally use reanalysis or climate model data as the driving hazard, along with varying exposure and vulnerability data to quantify these rare risks. Catastrophe models must be able to quantify losses at long return periods, for example at the 200-year return level in order to comply with the Solvency II directive[1]. Complying with this solvency directive becomes more difficult as there are only a few decades of coherent observational data from the historical period. Therefore, it is paramount to be able to understand and accurately quantify the risk of an event that would be at (or exceed) the 200-year return level. These risk estimates are often not freely available as catastrophe models are often licensed products with undisclosed vulnerability curves and calibration methods. Therefore, there is a clear need for a simple, transparent method to quantify high return levels associated with European windstorms for the wider risk community.

Projected extremes are dependent on the underlying distribution and it is therefore important to understand the total variability, or sampling, of extreme events in a historical period (Woo, 2019). Research on extreme precipitation has shown an increase in the intensity of potential extremes when considering a larger sample sizes (Thompson et al., 2017). This is particularly important for European windstorms due to the pronounced decadal variability in storm numbers (Donat et al., 2011b; Dawkins et al., 2016; Cusack, 2022) and losses (Klawa and Ulbrich, 2003). With a varying length of historical record this may result in particularly stormy periods, such as the early 1990's, having a stronger influence on projected risks. Shorter records that do not cover this period would likely underestimate the risk potential due to the absence of this key component of historical variability. Consequently, a key question that arises is how variable estimations of extreme gusts are with differing length historical records.

A well known driver of the frequency and strength of cyclones and windstorms in Europe is the North Atlantic Oscillation (NAO; Hurrell et al., 2003). The NAO drives these variations through modulations of the large-scale pressure gradient across the North Atlantic ocean, with more positive NAO phases leading to more cyclones with higher intensities than neutral or negative phases (Pinto et al., 2009; Gómara et al., 2014; Dawkins et al., 2016). It should therefore be expected that the NAO would modulate extreme gusts across Europe, although the extent of this has currently not been explored.

Consequently, the four main questions to be addressed in this study are as follows:

1. Can 200-year return level gust speeds from European windstorms be reliably estimated using openly-available observed windstorm footprints?

2. How important is the NAO in modulating high return level gust speeds?

3. Does the length of the historical record and choice of period influence return level estimates?

4. How could expected changes in the NAO under climate change lead to changing return levels?

---

[1]Losses must be covered with 99.5% confidence; European Union Solvency II Directive 2009/138/EC, available at https://eur-lex.europa.eu/legal-content/EN/ALL/?uri=CELEX:32009L0138





## 2 Data

### 2.1 Footprint Database

For analysing historical windstorms, a set of spatially coherent and validated footprints are required at high spatial resolution. Windstorm footprints are summaries of the overall hazard and are maps of maximum wind gust speed over a period of 72 hours. The Windstorm Information Service (WISC; WISC, 2017) project provided a set of footprints based on historical events for analysing the range, severity, and impact of historical windstorms (e.g. Koks and Haer., 2020; Welker et al., 2021). Footprints were produced by the UK Met Office through the downscaling of ERA-Interim and ERA-20C reanalysis data. The ERA-20C footprints range from 1940-2010 and ERA-Interim footprints from 1979-2014. In this study, the ERA-20C footprints from 1950-1978 and the ERA-Interim footprints from 1979-2014 are used. Footprints are provided at 4.4 km resolution and provide maximum 3-second gusts at a height of 10-metres for a 72-hour period of each downscaled storm (WISC, 2017). The WISC footprints are similar to those provided by the XWS project (Roberts et al., 2014), but extend further back in time and downscaled to a higher spatial resolution.

The storms to be downscaled were selected from a set of objectively identified cyclone tracks (Hodges, 1994, 1995), with those chosen being of relevance to the insurance industry and also those exceeding a pre-determined intensity threshold (Steptoe, 2017). The time period of the cyclone track for downscaling is defined as $\pm 36$ hours of the maximum intensity of the storm, with the maximum intensity defined as the time of maximum 925-hPa wind speed over land. The downscaling is performed in two steps, firstly the reanalysis data acts as boundary conditions for a 12 km version of the Met Office Unified Model that is initialized on four consecutive days for 30 hours. Following this, the 12 km model acts as boundary conditions for the 4.4 km model. The four consecutive days of 4.4 km downscaled data are then concatenated and the relevant time period of the track extracted. The final footprint is then the maximum wind gust value at each gridpoint. A spatial Gaussian filter is applied to the footprint in order to eliminate spurious extremes generated in the downscaling. All the footprints have been validated against observations and results in 124 high-resolution windstorm footprints for the period 1950-2014 for use in this analysis.

### 2.2 NAO Data

The NAO data used in this study is obtained from the NOAA CPC[2] and is calculated utilising rotated principal component analysis as described in Barnston and Livezey (1987). Daily teleconnection indices are obtained using the rotated principal component analysis applied to monthly 500 mb geopotential height fields from 1950–2000. Daily standardised values for the period 1950–present are obtained using the mean and standard deviation of the 1950–2000 climatological values. NAO data is extracted at the time of the WISC footprints.

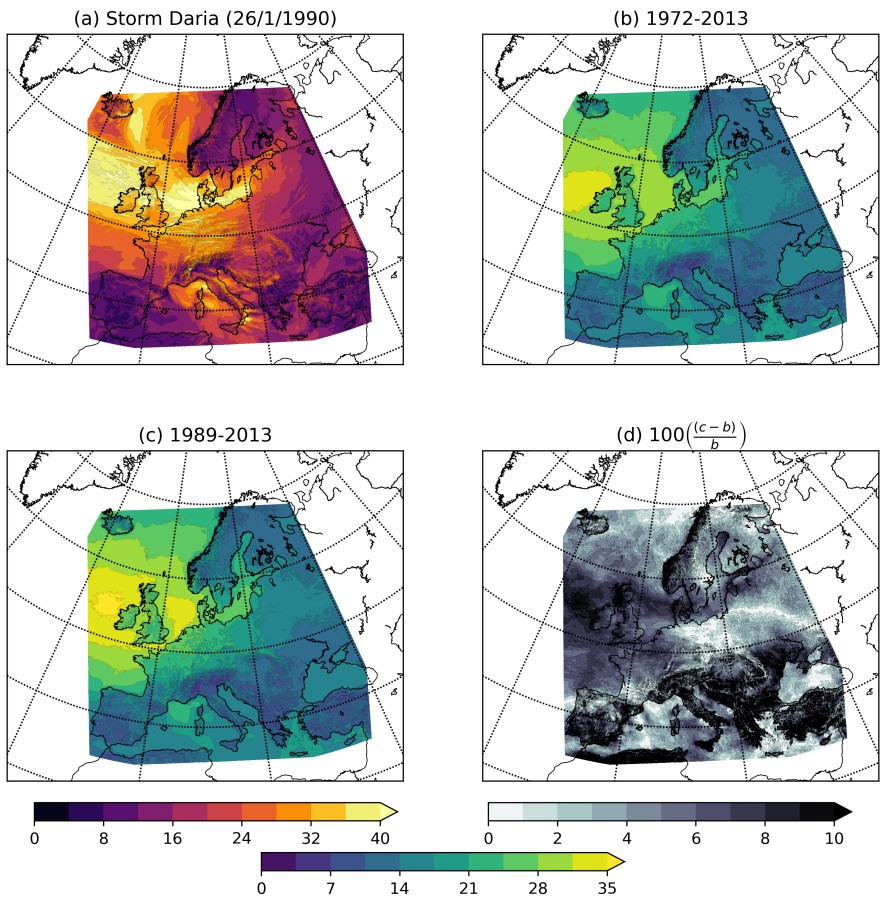

**Figure 1.** Storm footprints from the WISC catalogue. (a) Footprint of Storm Daria on 26/1/1990. Average footprint of two different length time periods (b) 1972-2013 and (c) 1989-2013. (d) The difference between the two periods as a percentage. Units are m s$^{-1}$.

## 2.3 Statistical Methodology

For building the statistical model to estimate high return period windstorm gusts across Europe, the return period of gusts from the WISC footprints (e.g. storm Daria; Fig, 1a) need to first be quantified. The return period is the expected time between successive events, and therefore is equal to the reciprocal of the rate of the process. Therefore, the formulation of the return period (in years), of a wind speed ($y$), for an event $Y > y$ is:

$$T(y) = \frac{1}{\lambda Pr(Y > y)} \tag{1}$$

[2]https://www.cpc.ncep.noaa.gov/products/precip/CWlink/pna/nao.shtml




where $\lambda$ is the rate of storm occurrence (per year). Windstorms can be separated in events with strong winds ($S$), and those without strong winds ($\bar{S}$). These events have different rates, $\lambda_S$ and $\lambda_{\bar{S}}$. For large return periods, it is assumed that the rate of occurrence for strong windstorms is considerably larger than for weak windstorms ($\lambda_{\bar{S}} Pr(Y > y \mid \bar{S}) \ll \lambda_S Pr(Y > y \mid S)$) and therefore $T(y) \approx \lambda_S Pr(Y > y \mid S)^{-1}$.

The exceedance probability can then be factored as follows:

$$Pr(Y > y \mid S) = Pr(Y > y \mid (Y > u) \cap S)\, Pr(Y > u \mid S) \tag{2}$$

where $u$ is a threshold that is large enough to use extreme value theory Generalized Pareto fits to the model $Pr(Y > y \mid (Y > u) \cap S)$. By assuming that the tail shape parameter is zero and in the Gumbel domain of attraction, this gives exponentially distributed excesses above the threshold:

$$Pr(Y > y \mid (Y > u) \cap S) = e^{\left(-\frac{y-u}{\sigma}\right)} \tag{3}$$

In equation 3, $\sigma$ is the tail scale parameter that can be estimated by Method of Moments from the mean excess above the threshold ($u$), yielding:

$$\hat{\sigma} = \frac{\sum_{y_i > u} y_i - u}{\sum_{y_i > u} 1} \tag{4}$$

Furthermore, the quantity $p(u) = Pr(Y > u \mid S)$ can be estimated from the relative frequency of exceedances:

$$\hat{p}(u) = \frac{1}{n} \sum_{y_i > u} 1 \tag{5}$$

and so if $u$ is taken to be the q'th empirical quantile ($u = y_q$), then $\hat{p}(u) = 1 - q$. As the WISC data is already a sub-selection of extreme wind speeds, a relatively low quantile ($q = 0.7$) is chosen to model the tail of the distribution. The sensitivity of the estimated return levels to the value of $u$ are discussed below.

Combining and rearranging equations 1–5 gives a prediction for the $T$-year return level:

$$\hat{y} = u + \hat{\sigma}\left(logT + log\hat{p}(u) + log\hat{\lambda}_S\right) \tag{6}$$

In order to test the functionality of the method, the statistical model is applied to the WISC data at three example locations. These three locations vary in latitude and represent varying influence from the leading pattern of variability, the NAO. The three locations are Bergen (60.4°N, 5.1°E), London (51.5°N, 0.4°W) and Madrid (40.4°N, 3.8°W). Figure 2 shows a demonstration





of the statistical model. The three locations have different gust speed distributions, with higher gusts at the northernmost

locations. The estimation of the 10-year return level at each location is very similar from our model and from the empirical

data. However, the 200-year return levels are larger than any gust from the WISC footprints at each location. For Bergen the

200-year estimate is 45.9 m s$^{-1}$ (39.7 m s$^{-1}$ WISC max), for London 39.0 m s$^{-1}$ (38.6 m s$^{-1}$), and for Madrid 30.1 m

s$^{-1}$ (24.1 m s$^{-1}$). The statistical model has also been applied to a set of windstorm footprints derived from the ERA5 re-

analysis and provided similar results (see Appendix A). Our model is therefore able to estimate high return period gusts that

are unprecedented in the observations.

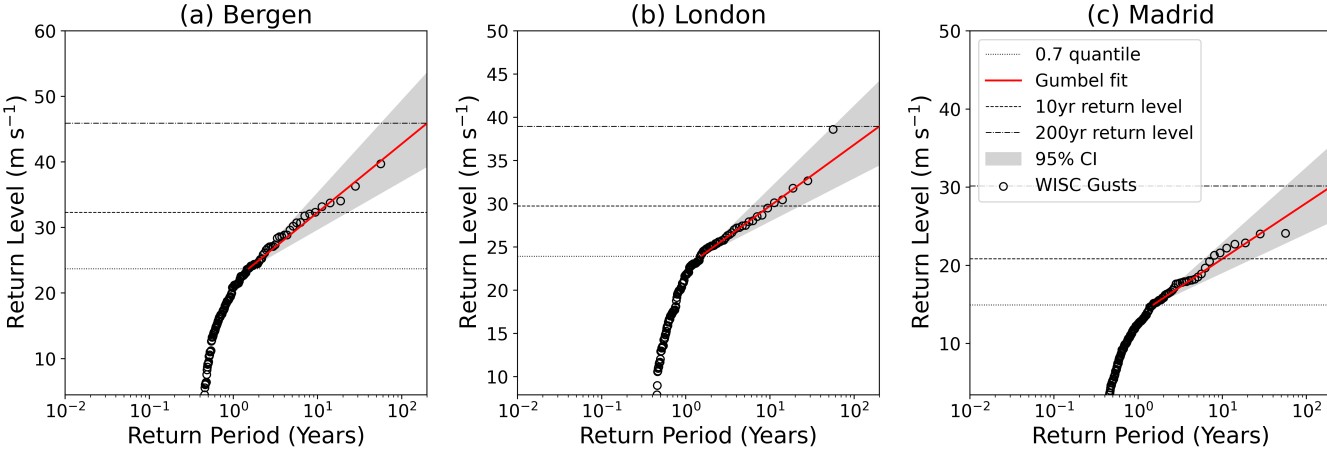

**Figure 2.** Tests of the Gumbel fit for high return periods at the 3 representative locations: (a) Bergen; 60.4°N, 5.1°E, (b) London; 51.5°N, 0.4°W, and (c) Madrid; 40.4°N, 3.8°W. The gusts for each location from 1950-2014 are plotted against their return period. Dotted black line indicates the 0.7 quantile, red line indicates the gumbel fit, black dashed line is the 10 year return period and black dash-dot line is the 200 year return period. Uncertainty estimates are calculated from the 95% confidence of the $\sigma$ parameter and are shown by the grey shaded region.

The chosen empirical quantile is 0.7 for estimating the extreme gusts. Figure S1a–c demonstrates how the 200-year return

level varies with a changing value of $u$ (or $q$). When $u$ is very small there are large variations in (and large values of) the

200-year return level due the inclusion of low gust speeds in our model fitting. There is a stabilisation in the 200-year return

level for $q \geq 0.7$, with variations within 5 m s$^{-1}$ for all three locations. This is the case up until the very highest thresholds,

where larger variations in the 200-year return level are again noted, however these high thresholds have a reduced number of

footprints in the model fitting, so as with the low thresholds, these are unlikely to provide realistic estimates of the high return

levels.



## 3  Results

### 3.1  Wind gust hazard maps

By applying this method to all gridpoints of the WISC footprints, estimates of the 200-year return levels can be produced for all of Europe (Fig. 3). The 10-year (Fig. 3a) and 200-year (Fig. 3b) return levels show a maximum in the return levels across NW Europe and the North Atlantic Ocean. This is expected as this is the region of strongest gust speeds in the WISC footprints (Fig. 1b,c). Return levels decrease radially from the UK, with lower values across Iberia, Italy, eastern Scandinavia, and eastern/south-eastern Europe. These lower values are associated with a lower frequency, and reduced intensity, of windstorms in the historical catalogue due to a greater distance from the main North Atlantic storm track (Priestley et al., 2020). The largest 200-year return levels exceed 60 m s$^{-1}$ to the northwest of Ireland. Over land, the largest 200-year return levels are 55-60 m s$^{-1}$ across eastern Scotland, northeastern England, and Denmark, with values above 40 m s$^{-1}$ for the majority of the rest of the British Isles, northern France, Benelux and Germany (Figure 3b).

The dominant parameters in the return level estimates are the threshold ($u$; Fig. 3c), and the mean excess ($\hat{\sigma}$; Fig. 3d). The spatial pattern of $u$ (Fig. 3c) is similar to the return level maps (Fig. 3a,b) in that is features a maximum in the northwest of the domain in the area surrounding the North Atlantic. The magnitude of $u$ decreases toward southern and eastern Europe. This pattern suggests a strong NAO influence, with larger values in the region where the NAO drives high wind speeds and damaging windstorms (Pinto et al., 2009). Unlike $u$, $\hat{\sigma}$ does not have the same N-S dipole over western Europe (Fig. 3d). Instead values vary much less, with only a 4-5 m s$^{-1}$ variation across the entire WISC domain. The variation in $\hat{\sigma}$ is not coherent and is therefore consistent with this component being independent of large scale patterns, such as the NAO.

### 3.2  Generalizing the model to include an NAO covariate

The parameter $u$ displays a pattern that is indicative of an NAO influence, and can therefore be generalized to include such variations. If the threshold is chosen to be the q'th empirical quantile then $u = \hat{y}_q$, then $\hat{y}_q$ and $\sigma$ can be generalized to be a function of a covariate $x$. The covariate $x$ will be the daily NAO index at the time of a WISC footprint occurrence. By estimating the threshold using linear quantile regression, and $\sigma$ is estimated using generalized linear regression for a Gamma distribution with identity link, this yields:

$$\hat{y}_q = \hat{\beta}_0 + \hat{\beta}_1 x \tag{7}$$

$$\sigma = \hat{\alpha}_0 + \hat{\alpha}_1 x \tag{8}$$

Applying these regressions to the WISC data yields the parameters shown in Figure 4. The $\hat{\beta}$ parameters (Fig. 4a,b) exhibit the typical NAO-influence pattern, with larger values and positive regression coefficients across the northwest of the WISC domain and smaller values/negative coefficients across eastern, southeastern, and southern Europe. This indicates that more

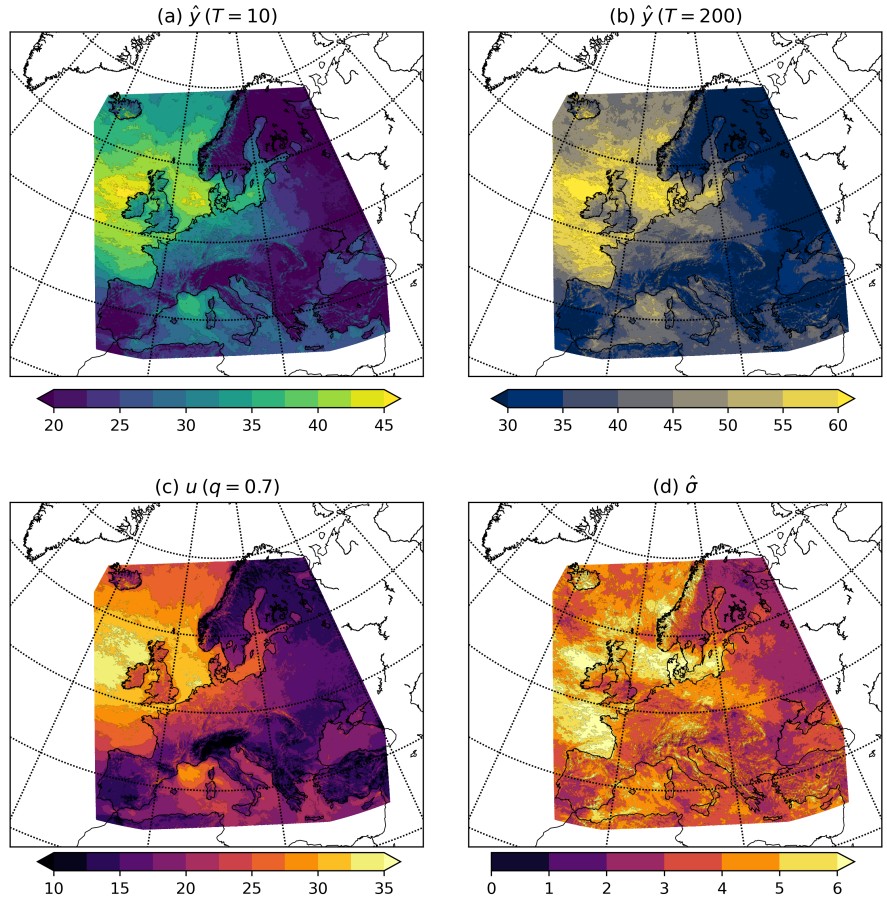

**Figure 3.** Hazard maps generated by the statistical model for the (a) 10-year and (b) 200-year return levels across Europe. (c) The 0.7 quantile threshold used to calculate the return levels. (d) The mean excess (sigma) used in the return level calculations. All units are m s$^{-1}$.

northerly latitudes and NW European locations have a positive NAO/gust threshold relationship and vice versa. The $\hat{\alpha}$ param-
eters (Fig. 4c,d) do not feature an NAO influence and values are more variable across all of Europe. The N-S dipole is not
evident and it therefore appears that the NAO has minimal influence on the windstorm gust excesses. Despite this finding for
the $\hat{\alpha}$ parameters, it may be that an NAO relationship is undetectable in our small data sample and that with a larger pool
of windstorm footprints a signal may emerge. Due to the undetectable signal in the $\hat{\alpha}$ parameters the NAO covariate is only
applied to the threshold ($u$) in the statistical model, and therefore equation 6 becomes:

$$\hat{y} = \hat{\beta}_0 + \hat{\beta}_1 x + \hat{\sigma}\left(logT + log(1-q) + log\hat{\lambda}_S\right) \tag{9}$$

Figure 5 shows the NAO covariate model (equation 9) tested at the same three gridpoints as previously. Each location
features a different NAO relationship. The most northerly point, Bergen, has a strong positive relationship between return level


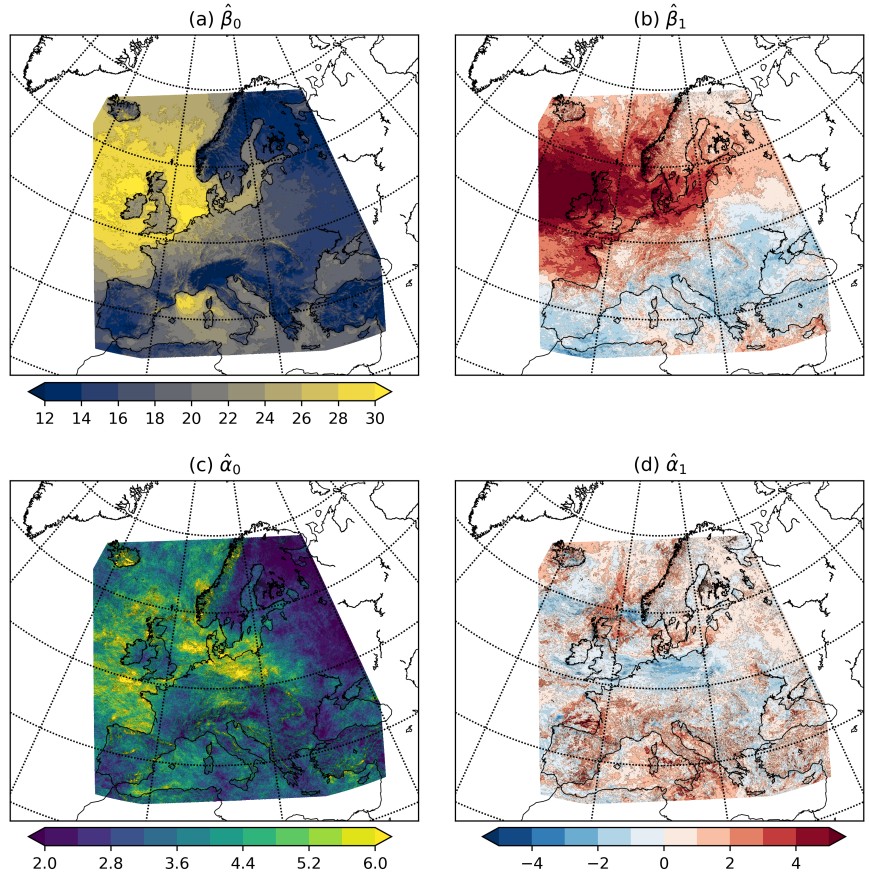

**Figure 4.** Maps of the (a) $\hat{\beta}_0$, (b) $\hat{\beta}_1$, (c) $\hat{\alpha}_0$, and (d) $\hat{\alpha}_1$ parameters for regressions applied to the WISC footprints from 1950–2014 with daily NAO covariate. Units are (a,c) m s$^{-1}$ and (b,d) m s$^{-1}$ st. dev $^{-1}$.

and the NAO. An NAO of +1.5 standard deviations yields a 200-year return level of >50 m s$^{-1}$. For the London gridpoint, this NAO relationship is still positive, although weaker than for Bergen, as would be expected. Finally, for the Madrid gridpoint a

negative/neutral relationship is present, and therefore when the NAO is more positive, the estimated return level gets smaller. The reduced NAO sensitivity for the Madrid gridpoint is to be expected from the in-land and southerly nature of this location.

     Applying the NAO covariate method to all gridpoints yields the return levels for the 10 and 200 year return periods shown in Figure 6. The same regional variation and distribution in return levels across Europe is seen as in Fig. 3. Return levels are higher for the 200-year return period (Fig. 6b) than the 10-year (Fig. 6a) with values ∼10 m s$^{-1}$ higher across northwestern

Europe. It should be noted that the calculated 200-year return levels are higher when using the NAO covariate (Fig. 6b, 3b), with maximum values of >65 m s$^{-1}$ over the North Sea and North Atlantic Ocean, and values exceeding 50 m s$^{-1}$ across the northern British Isles, Denmark, and northern Germany. The variability in return level with an NAO state of +0.5 (Fig. 6a,b) and -0.5 (Fig. 6c,d) is also apparent for both return levels, with higher values across the NW of the domain for the NAO+ state.





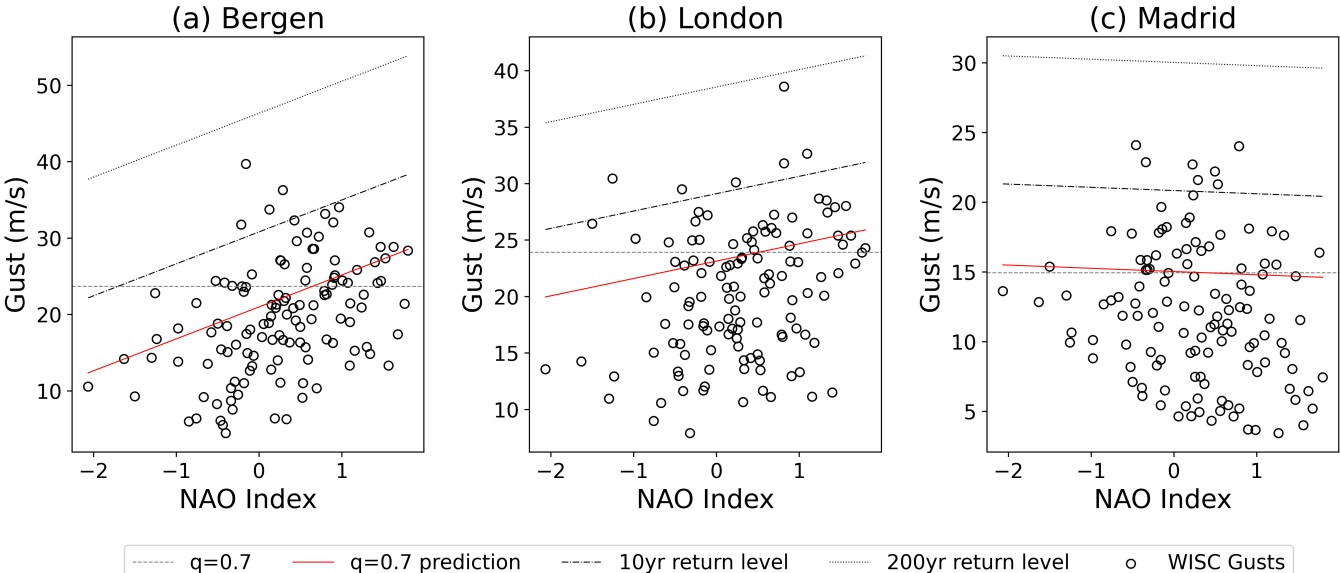

**Figure 5.** Dependence of gusts on the NAO at the 3 representative gridpoints: (a) Bergen; 60.4°N, 5.1°E, (b) London; 51.5°N, 0.4°W, and (c) Madrid; 40.4°N, 3.8°W. Scatter points are the WISC gridpoint gusts against the daily NAO index. Horizontal dashed black line indicates the 0.7 quantile for each location. The red line is the q=0.7 estimation based on the NAO-gust regression. Dash-dotted black line indicates the 10-year return level, and the dotted black line is the 200-year return level using the Gumbel domain estimation.

The spatial pattern of the 10-year and 200-year return levels (Fig. 6) differs due to the differing contribution of $u$ and the excesses ($\hat{\sigma}(logT)$) with return period (Fig. S2). The contribution of the NAO-dependent $u$ decreases with increasing return period and the contribution of the NAO-independent $\hat{\sigma}(logT)$ increases. Therefore, as the influence of the NAO is only detectable in the regression of $u$, it can be said that the relative importance of the NAO on our estimated return levels decreases with an increasing return period.

### 3.3 Sensitivity to choice of historical period

One factor that can contribute to the estimation of the 200-year return level is the length of the historical catalogue. The varying length of catalogue has a substantial impact on the average footprint (Fig. 1b–d) and when applied to the statistical model at a return period of 200 years, these differences are likely to be amplified. The 200-year return level estimates vary a lot when using different length historical catalogues (Fig. S3). The mean squared error (MSE) of the 200-year return level for historical catalogues from a year in range of 1951-2014 relative to the full catalogue of 1950-2014 increases with shorter catalogue lengths (Fig. S3a) with gridpoint return level estimates being both under or over-estimated dependent on the historical catalogue used (Fig. S3b,c).

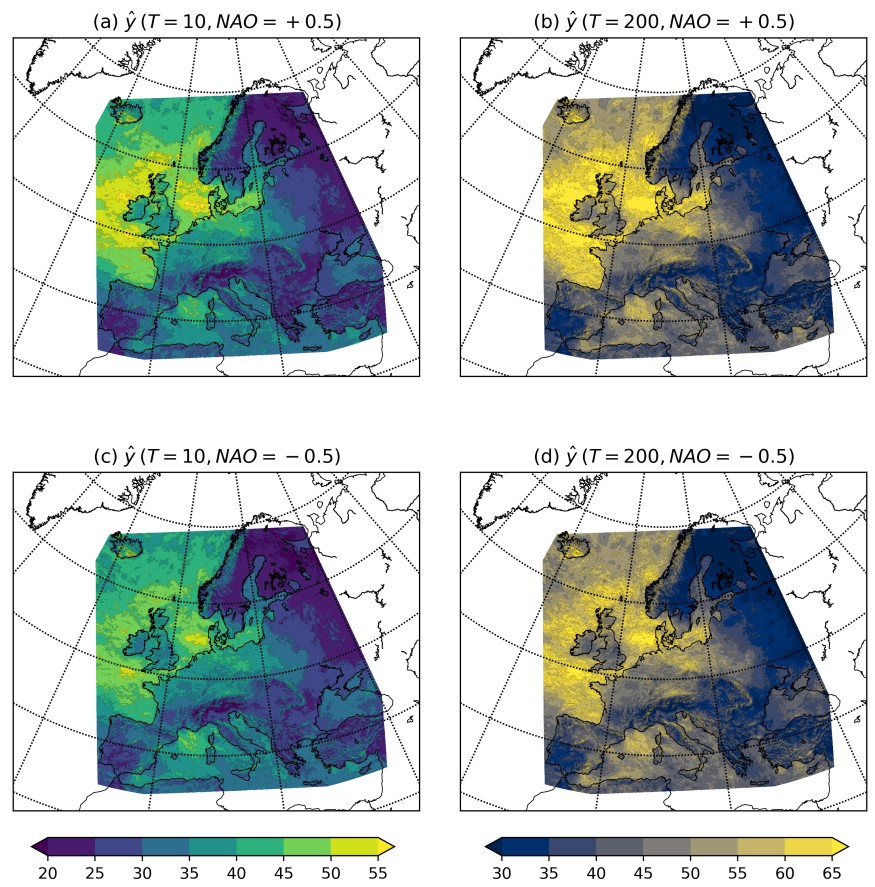

**Figure 6.** NAO dependence of return levels. Maps of the (a,c) 10-year and (b,d) 200-year return level across Europe using the NAO covariate. Maps are shown for an NAO signal of (a,b) +0.5 and (c,d) -0.5. Units are m s$^{-1}$.

In order to quantify how different catalogue lengths compare in their return level estimates, an independent time period is required for validation. For this, a 5-year time period is taken (e.g. 01/2010-12/2014) and the 200-year return levels are esti-

mated. The historical catalogues for the preceding years starting at a length of 1 year and extending all the way out to 60 years (i.e. 2009, 2008-2009, 2007-2009,..., 1950-2009 in this example case) are taken and return levels estimated. The MSE of all catalogue return levels are then calculated relative to the 5-year validation period to quantify of the optimal historical catalogue length required to minimise the error in estimated return levels. To account for natural variability all possible 5-year validation periods are tested with start years ranging from 1955 to 2010.


Applying this methodology to the same three gridpoints noted above we find that all locations (Fig. 7) feature a high MSE for the shortest catalogue lengths for the 200-year return level. At Bergen (Fig. 7a) this is less evident in the median, but large values of MSE are notable in the spread, which reduces after a catalogue length of 2-3 years. The high MSE with short





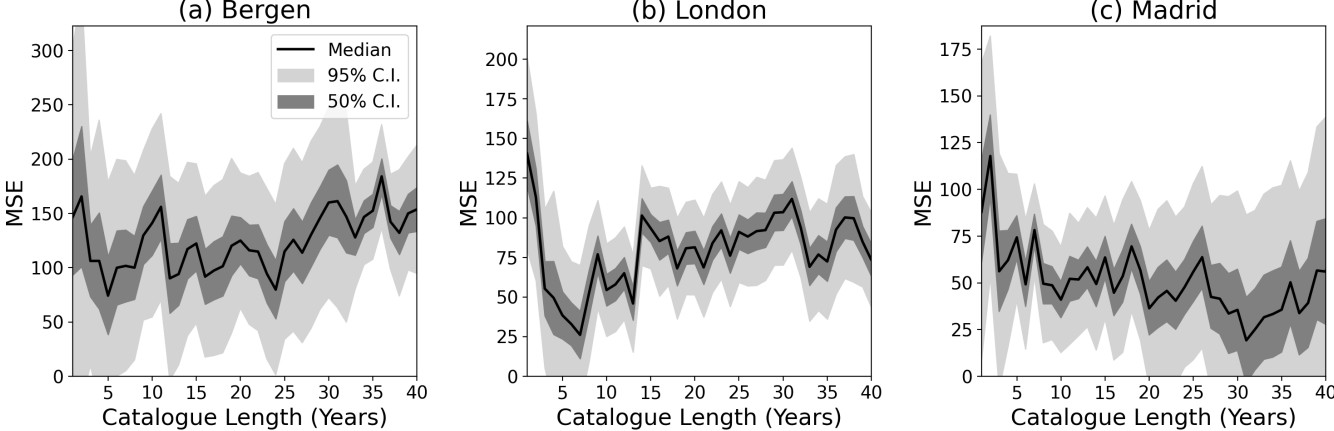

**Figure 7.** Mean square error of the 200-year return level estimation of different length historical catalogues against a subsequent 5 year period from the WISC catalogue for (a) Bergen, (b) London, and (c) Madrid. Solid black line shows the median mean squared error when using all possible periods. The dark and light gray areas represent the 50% and 95% confidence interval on the standard error respectively.

catalogue length is expected due to the limited sample of footprints contributing to the return level estimate. Increasing cata-

logue lengths to above 2 years causes a reduction in the MSE at all locations, with limited improvements after ∼5-15 years. Therefore, historical catalogues longer than 5-15 years do not yield improvements in the return level estimation at these three locations. Adding uncertainty is the variations in the MSE estimate from year-to-year, and large spread in the MSE. With a limited historical catalogue, potential reductions in uncertainty or MSE for catalogue lengths longer than 40 years cannot be quantified.

**3.4   Simulating the NAO for a more robust catalogue length estimation**

For a more confident estimation of the optimal catalogue length, a timeseries of the NAO and the windstorm model parameters are simulated. The NAO is simulated as a sinusoidal variation of period 70 years (e.g. Wanner et al., 2001; Olsen et al., 2012; Wang et al., 2017) and amplitude of 1.5 to match the variations from the WISC footprints (not shown). The WISC footprints occur at a rate of ∼2.1 per year and the model is built to fit the top 30% of these events, therefore, occurrences of extreme

storms and their NAO phase can be estimated from these frequencies. This entire simulation is done for 1000 years (Fig. 8a).

Using the regressional relationships between the WISC footprints and the NAO state (Fig. 4a,b), the value of $u$ (Fig. 8a) and $\sigma$ can be estimated in the simulated timeseries and determine the resultant gust (Fig. 8b). Comparing the WISC gridpoint gusts for Bergen with our simulation (Fig. 5a, 8c) it is notable that the simulated gusts exceed those in the WISC dataset

considerably, with gusts in excess of 60 m s$^{-1}$.

Using this simulated timeseries, the 10-year and 200-year return levels can be estimated for different catalogue lengths for our three locations, with analysis performed as in Figure 7. At Bergen the highest MSE is seen for the shortest catalogue lengths

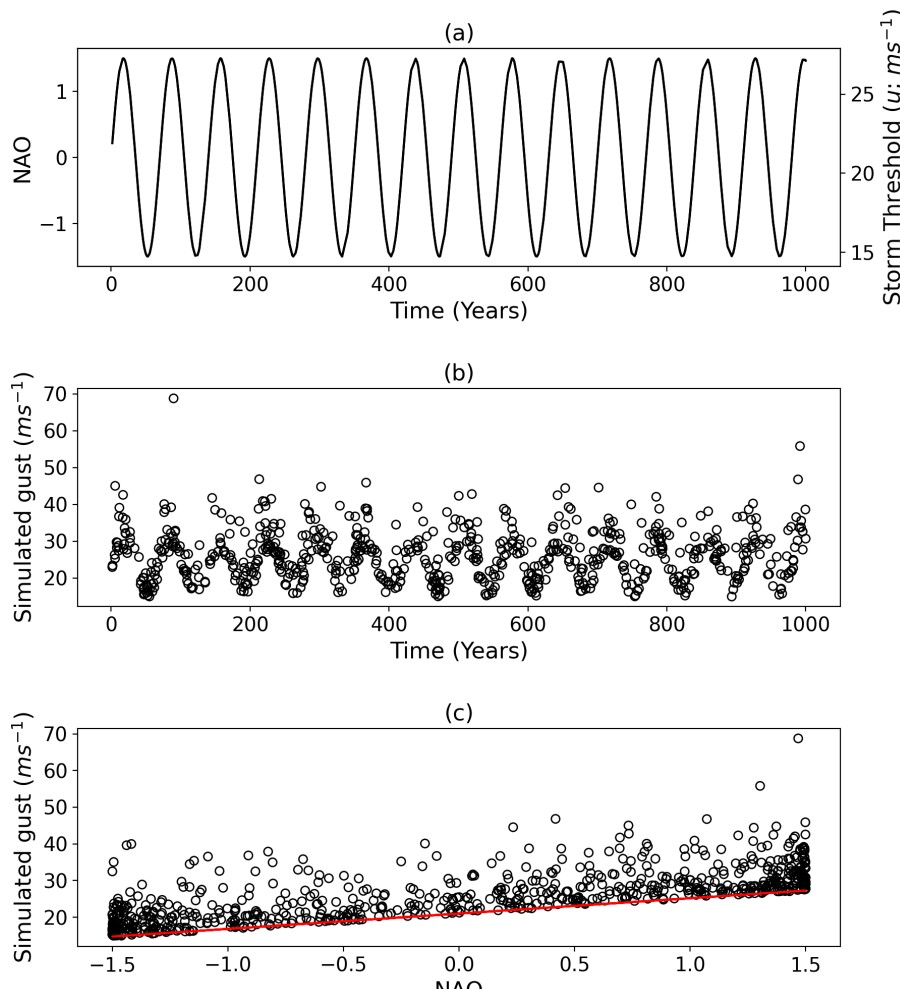

**Figure 8.** Summary of simulated events at Bergen based on 1000-year NAO timeseries. (a) NAO phase and simulated storm threshold ($u$) at time of simulated events. (b) Simulated event gusts (m s$^{-1}$). (c) NAO phase plotted against the simulated gusts. Red solid line shows the $q = 0.7$ regression line.

(Fig. 9). As in Figure 7a), this is a result of the large inter-annual variability and the chance that one year will be representative of the following 5 being very small. For the 200-year return levels (Fig. 9b), a large reduction in the MSE is then seen as catalogue length increases to 20 years. For catalogue lengths longer than this the MSE changes are small. Unlike the WISC data, there is now little variation in the MSE out to catalogue lengths in excess of 250 years and a much reduced spread and variation in the median. Therefore, there is limited benefit to using historical catalogues longer than ~20 years when estimating the 200-year return level.




At the 10-year return level (Fig. 9a) a reduction in MSE occurs up to a catalogue length of ∼10 years. Unlike the 200-year
MSE, there is an oscillation in the MSE that has a period of ∼70 years and results in a maximum MSE for catalogue lengths of
40 years and a minimum after ∼80 years. The ∼70 year oscillation is consistent with the defined periodicity of the NAO sim-
ulation. The oscillation is present as the NAO influence is more pronounced for shorter return periods (Fig. S2). Consequently,
the 10-year return levels more strongly reflect the variation in NAO, whereas the 200-year return levels will not. To reduce
the MSE, the NAO signal of the historical catalogue relative to the 5-year validation period is important. For shorter catalogue
lengths the NAO signal (and simulated gusts) will be similar, however, catalogue lengths of ∼35 years feature an NAO signal
that is most different from the 5-year validation period and hence the MSE will be most different. When a full NAO cycle is
sampled (∼70 years), the full proportion of NAO variability is captured and there is a minimum in MSE. For catalogue lengths
longer than the prescribed NAO periodicity, the average NAO signal varies less and a dampening of the MSE variation is seen.

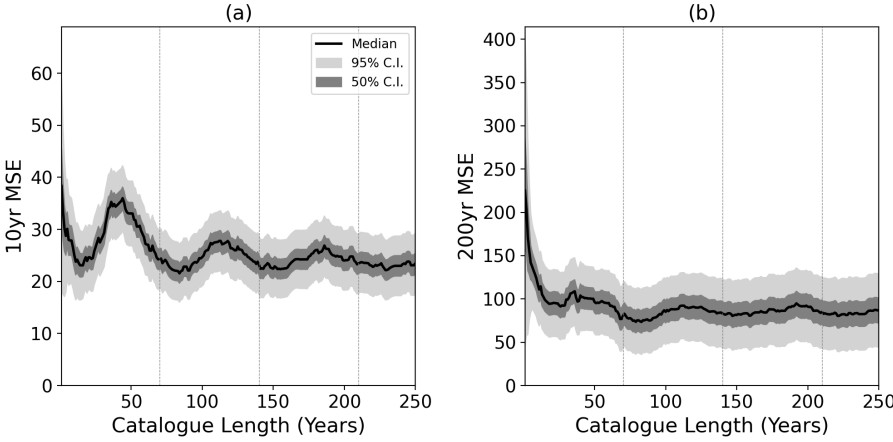

**Figure 9.** Mean square error of the (a) 10-year and (b) 200-year return level estimation of different length historical catalogues against a
subsequent 5 year period with events from a 1000-year simulation for Bergen. Solid black line shows the median MSE from all possible
periods. The dark and light gray areas represent the 50% and 95% confidence interval on the standard error respectively. Vertical dashed grey
lines indicate the periodicity of the NAO used in simulations.

The structure of MSE evolution seen at Bergen (Fig. 9) is similar at London (Fig. S4) and Madrid (Fig. S5). For the 10-year
return levels the oscillatory evolution at Bergen (Fig. 9a) is less apparent due to the weaker influence of the NAO on return
levels (Fig. 4, 5). As a result it is less important to sample a full cycle of the NAO and the MSE varies little for catalogue
lengths longer than ∼20 years.

## 3.5 Change in return levels from plausible future NAO states

Recent estimates of the historical trend of the NAO has been upwards at a rate of +0.15 standard deviations per decade (1950–
2020; Blackport and Fyfe, 2022). If it is assumed that the historical averaged NAO state is neutral, then a state of +1.5 standard
deviations in 100 years is reasonable. For estimating return levels of gusts in this future NAO state, events are simulated as

in section 3.4. To account for the natural variability of NAO states, a 70-year period is used that features two 35 year periods

at +(-) 0.5 standard deviations than the reference state. For the historical state the reference is NAO neutral and events are simulated at -0.5 and +0.5. For the future state the reference NAO is +1.5 and events are simulated at +1 and +2. The 70 year periods are simulated 100,000 times for our three gridpoints to obtain robust return levels. Selecting a model threshold and estimating the return levels of this two component model differs slightly from the earlier methodology and is documented in appendix B.

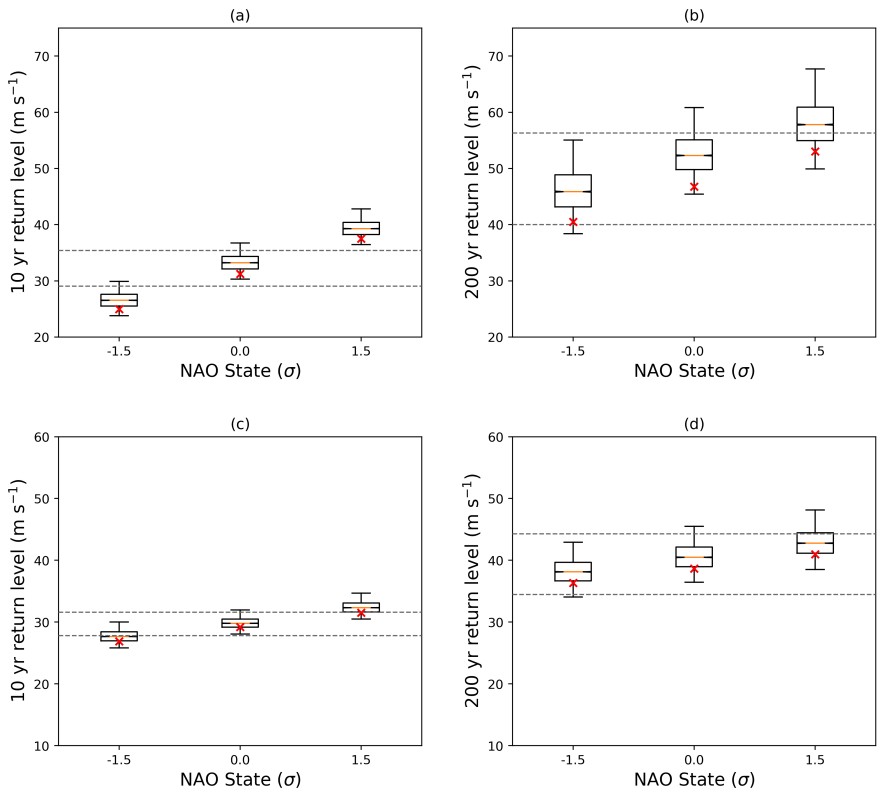

**Figure 10.** Boxplots of estimated (a,c) 10-year and (b,d) 200-year return levels for (a,b) Bergen and (c,d) London based upon simulated gusts. Return levels are estimated from 100,000 70-year simulations with NAO phase that varies to $\pm0.5\sigma$ of a set NAO state. Boxplots show the median return level of these 100,000 simulations and boxes extend to the 25th and 75th percentiles, with whiskers extending to the 2.5th and 97.5th percentiles. Red crosses indicate the theoretical return level based upon a two-state NAO. Dashed horizontal grey lines indicate the 95% confidence interval of the WISC 10-year and 200-year return levels.

The estimated return levels at Bergen for the NAO state of +1.5 are largely outside the WISC uncertainty for the 200-year

return level (Fig. 10b). Therefore, if historical trends were to continue for the next 100 years then the 200-year return levels would likely be unprecedented. This is more evident at the 10-year return level (Fig. 10a) due to the increased influence of the NAO on these shorter return periods. For periods of extreme negative NAO states the reverse is seen, with a reduction in the


return levels to the lower end of the historical estimation. For all NAO states the return levels estimated using the simulated events are positively biased relative to theoretical values using exact parameters (Fig. 10). This is a result of the additional

natural variability present in the simulated events from the varying NAO state (see appendix B). Therefore, models that assume a stationary historical climate are also likely to be positively biased relative to actual return levels.

Similar changes in return levels are also seen for the future NAO state at London (Fig. 10c,d). However, as the NAO has a reduced influence the increase in 200-year return levels is less apparent with most estimates within the historical uncertainty

(Fig. 10c). Nevertheless, the potential increases in 10-year return levels are still large enough to be greater than the historical uncertainty. At Madrid (Fig. S6), the NAO influence is so small that any changes in the return levels are insignificant compared to the historical uncertainty.

## 4   Conclusions

This study has presented a method for estimating high return period gusts for the European domain based upon a set of

observed windstorm footprints. This simple and transparent method operates without requiring the complexities of a full catastrophe models and is agnostic to the input dataset. The amount of historical data required to minimise errors in the return level estimation has been tested, and the sensitivity of return levels to potential future NAO states has been quantified. The key conclusions are as follows:

–   Return levels at the 200-year return period have been quantified and associated uncertainties quantified. Return levels are

higher over northwestern Europe, with the NAO playing a role in increasing (decreasing) return levels across northern (southern) Europe.

–   The NAO is important for setting the threshold for gust speed return levels, but the largest gust speeds feature stochastic noise. The NAO modulates the lower return levels and is key in determining the tail location parameter. The tail scale parameter is much more variable with no detectable influence from the NAO.

–   Only 20 years of historical data is required to reduce errors in 200-year return level estimates. Increasing historical records longer than this does not lead to meaningful improvements in return level estimates. Estimates of 10-year return levels are more sensitive to the length of historical catalogue due to the larger NAO influence.

–   Under future NAO states the 10-year and 200-year return levels are likely to be unprecedented relative to historical values. Return level estimates are outside the historical uncertainty across northern Europe, with minimal changes across

southern Europe. This would lead to considerably higher gusts and impacts than those currently considered.

Our modelling approach makes several simplifying assumptions. Firstly, for estimating the tail a zero shape parameter is used. This implies that an infinite return level is possible for the longest return periods. This is inaccurate and in reality the shape parameter would be negative as there are frictional processes, instabilities, and limits to the amount of transferrable



kinetic energy from a windstorm. However, we are confident in our estimations as the generated return curves are a good fit for the input data. These results provide additional data points to compliment estimations made from more complex catastrophe models that will help guide risk modellers and aid understanding of potential European windstorm risks.

With regards to the data used, there is a historical limit due to the constraints of observations of severe European storms. This limit to 1950 may result in considerably large events that would constrain the model being missed, or key components of the natural variability being excluded. However, with ongoing efforts to accurately represent historical storms (Hawkins et al., 2022), this may be improved in the future. Alongside this, the data used from the WISC project is only a small sample of all windstorms over Europe representing just the most intense events and this may lead to a skewing of our return level estimations toward the worst case scenario.

Using plausible future NAO states in our methodology to estimate return levels only represents the changes the circulation aspects of the NAO. One factor not considered in the large increase in return level for northern latitudes of Europe is the thermodynamic contribution. As a large amount of the potential increase in cyclone wind speeds across Europe is a result of moist processes (Dolores-Tesillos et al., 2022; Binder et al., 2022), it is likely that gusts would increase further beyond those predicted in the model here. These changes could have further implications on potential losses. Only the NAO was considered as a low-frequency modulator of windstorm intensity in our model. However, other phenomena have been shown to influence European winter weather through connections to the NAO. These are phenomena such as tropical precipitation (Scaife et al., 2017, 2019), the Atlantic multi-decadal oscillation (Börgel et al., 2020), and the El Niño Southern Oscillation (Zhang et al., 2019a, b) to name a few. These low-frequency phenomena all have partial connection to the NAO, and therefore may play a role in modulating windstorm strength, either through the threshold or excesses in this model. Investigating potential linkages further may lead to improvements in the model and therefore more accurate return level estimations.

There are numerous future applications of our model. Firstly, this model could be used to investigate future changes in return levels. With projected future changes in the NAO, and other leading modes of European weather variability (Fabiano et al., 2021; Blackport and Fyfe, 2022), the simulation of the NAO can be modified to apply trends and changes in variability to determine how this would affect the higher return period gusts. This could provide vital insight to the projected end-of-century increase in European storminess (Donat et al., 2011a; Zappa et al., 2013; Priestley and Catto, 2022; Dolores-Tesillos et al., 2022). Finally, despite this model being developed for European windstorm risks, it could be easily adapted for other hazards/risks and act as a framework for assessing extreme events of other perils. Validation and testing would be required, but this model can act as an open-source and transparent method for investigating and quantifying a range of natural hazards.

*Code availability.* Model code is available at the request of the author.





*Data availability.* ERA5 reanalysis is available from the Copernicus Climate Change Service Climate Data Store (https://cds.climate.copernicus.eu/#!/search?text=ERA5&type=dataset). WISC footprints are available at the request of the author.

*Author contributions.* MP wrote the manuscript and performed all analysis. MP, DS, and AS devised the study and edited the manuscript. DS formulated the statistical methodology. DB provided feedback and comments on the manuscript.

*Competing interests.* The authors declare that they have no competing interests.

*Acknowledgements.* This research was funded and supported by the WTW Research Network. A.A.S. was also supported by the Met Office Hadley Centre Climate Programme funded by BEIS (Department for Business, Energy & Industrial Strategy) and Defra (Department for Environment, Food & Rural Affairs).




## Appendix A: Applying the statistical model to ERA5 footprints

In testing the applicability of the statistical model it has also been applied to a set of footprints from the ERA5 reanalysis (Hersbach et al., 2020). Footprints were created from a set of extratropical cyclone tracks. Tracks were identified following the method of Hodges (1994, 1995), which uses 850 hPa relative vorticity as a tracking variable. Tracks were created from 1-hourly data for the entire calendar year (1st Jan – 31st Dec) for the period of 1980-2020. Full details of the cyclone identification and tracking method can be found in Hoskins and Hodges (2002). Tracks were filtered to those passing through a defined North
Atlantic/European domain (30°N-75°N, 40°W-40°E) to encapsulate all tracks that would pass through continental Europe and to cover all the WISC domain.

Footprints were created using hourly near-surface (10-metre) wind gusts. This data provides the maximum 3-second wind gust at each gridpoint in the previous hour. At each track timestep the gusts in the surrounding 5° are associated with the storm
and the maximum gust at each gridpoint across the lifecycle of the track is retained in order to create a coherent footprint. This method closely follows that of Roberts et al. (2014). An example footprint is shown in Figure A1.

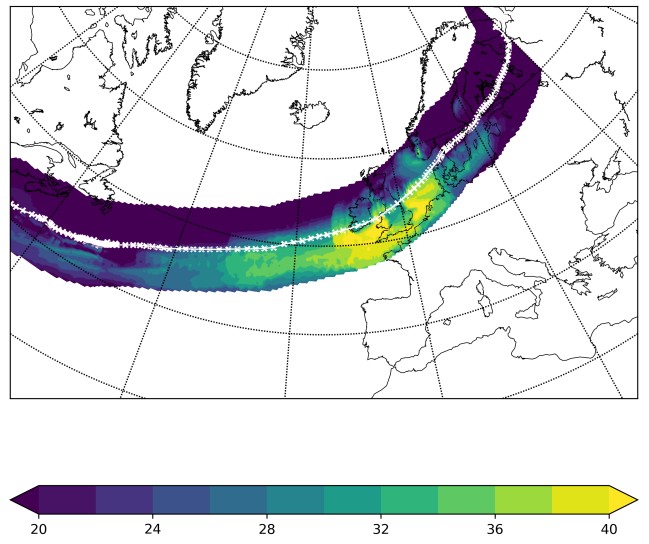

**Figure A1.** Wind gust footprint from ERA5 of track forming on 20/1/1990 and dissipating on 28/1/1990. Coloured shading represents the maximum gridpoint wind gust in m s$^{-1}$. White line and crosses indicates the track and hourly track points respectively.

These footprints are then applied to the statistical model described in section 2.3 and perform analysis as in Figure 2. As there are many more footprints in our ERA5 dataset than from WISC, there are a considerably greater number of points to





fit our model (Fig. A2a–c). As a result of the increase in the number of footprints, which are mainly of lower gust speeds, a
higher threshold quantile for ERA5 is used than for the WISC footprints, with $q = 0.9$ for the ERA5 footprints. Despite the
increase in the number of footprints, the higher value of $q$, and the coarser resolution of the ERA5 data, the model fit is a good
approximation for the data and estimates of the 200-year return levels are able to be made.

The estimated 200-year return levels (Fig. A2a–c) are different from those estimated from WISC (Fig. 2a–c), as would be
expected due to the differing input data. However, the values are consistent with those from WISC, with the main difference
being an increase in the value of the 200-year return level for the London gridpoint (Fig. A2b). What is further re-assuring
about the applicability of our model is the stability of the 10-year and 200-year return levels for high quantiles used in the
model (Fig. A2d–f). Estimations of these two return levels varies by less than 5 m s$^{-1}$ for $q \geq 0.9$.

**Figure A2.** (a–c) As Figure 2 but using ERA5 footprints as an input. (d–f) The sensitivity of the 10-year (black solid) 200-year (red solid)
return level to the choice of quantile threshold for (d) Bergen, (e) London, and (f) Madrid. Vertical black lines indicate the 0.5 and 0.9
quantiles. Thin red and black lines represent the 95% confidence interval based upon the estimate of the $\sigma$ parameter.




## Appendix B: The role of natural variability

To gain insight into the effect of natural variations in modulators of the extremes, it is useful to consider a simple two component mixture model. Consider a hazard process that is an equal mixture of two processes $Y_a(t)$ and $Y_b(t)$ that have exponential exceedances above thresholds $a$ and $b > a$ respectively. Then for $y \geq b$,

$$
\begin{aligned}
\Pr(Y > y) &= 0.5\Pr(Y_a > y | Y > a)\Pr(Y_a > a) + 0.5\Pr(Y_b > y | Y > b)\Pr(Y_b > b) \\
&= 0.5\exp(-(y-a)/\sigma)\Pr(Y_a > a) + 0.5\exp(-(y-b)/\sigma)\Pr(Y_b > b)
\end{aligned}
$$

If thresholds $a$ and $b$ are chosen to be the q'th quantiles of $Y_a$ and $Y_b$ then

$$
\begin{aligned}
\Pr(Y > y) &= 0.5\exp(-(y-a)/\sigma)(1-q) + 0.5\exp(-(y-b)/\sigma)(1-q) \\
&= 0.5(1-q)\left(e^{a/\sigma} + e^{b/\sigma}\right)e^{-y/\sigma}
\end{aligned}
$$

Hence, the T-year return level is given by

$$
y_T = \sigma \log\left(0.5(e^{a/\sigma} + e^{b/\sigma})\right) + \sigma\left(\log T + 1 - q + \log \lambda_S\right)
$$

which equals that derived in Eqn. (7) if one sets $u = \sigma \log\left(0.5(e^{a/\sigma} + e^{b/\sigma})\right) \in [a,b]$. By integration of the p.d.f., it is straightforward to derive the mean excess of this process above threshold $u \in [a,b]$:

$$
E(Y - u | Y > u) = \sigma + \frac{b-u}{e^{-(u-a)/\sigma} + 1}
$$

which can be seen to be greater or equal to $\sigma$. Hence, estimating $\sigma$ by assuming it is equal to the mean excess will lead to a positive bias that vanishes only as $u \to b$. In summary, the effect of natural variability in the tail location parameter is to cause
the mean excess to overestimate the tail scale parameter, which in turn will lead to overestimates of return values for longer return periods. Put simply, if natural variability is not taken into account, the magnitudes of the extreme events will appear to be more variable than they actually are.



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
