# Peer review of "Return levels of extreme European windstorms, their dependency on the NAO, and potential future risks"

_Natural Hazards and Earth System Sciences, 2023_

## Author Comment (AC1)

Reviewer response to manuscript nhess-2023-22 'Return levels of extreme European windstorms, their dependency on the NAO, and potential future risks' by Matthew D. K. Priestley, David B. Stephenson, Adam A. Scaife, Daniel Bannister, Christopher J. T. Allen, and David Wilkie

**Reviewer 1**

The manuscript presents an interesting exploration of European windstorm extremes, how these can be statistically modelled, and their dependence on NAO. The manuscript poses several useful and relevant questions, motivated by identified knowledge gaps. I think overall the manuscript is well structured and provides some useful insights into the answers to the questions posed. I appreciate the discussions of the limitations of the analysis but think in some places the methods could be better justified to strengthen the credibility of the conclusions made. I would thus recommend the manuscript for publication after moderate/major revisions. Below are some suggestions for improvement which I hope the authors may consider in preparing a final version.

We thank the reviewer for their comments and consideration of our manuscript. We have taken on board all comments and believe our manuscript is now much stronger as a result. In particular, we have slightly altered some of the analysis around Figures 7 and 9 (sections 3.3 and 3.4). We now use a 10-year validation period for testing our return level estimates and this makes our results considerably clearer, but importantly there are no major changes to the conclusions we made in our initial submission. Below we detail all our responses to the comments and state the changes we have made in the text.

**MAIN POINTS**

- Sections 3.1 and 3.2 state that the scale parameter is 'independent of large scale patterns such as the NAO' and there is an 'undetectable signal in the α parameters'. It seems that these statements are made based purely on a visual inspection of Figure 4. My concern is that this these are possibly strong statements without justification based on further exploration. Could it be possible that there is an NAO signal in some regions and not in others (e.g. over some parts of Northern Europe where $\alpha_1$ is ~-3)? Could the authors provide a test of significance for this relationship, at least for the three most studied locations (Bergen, London, Madrid)? This would give greater credibility to this assumption and the results that follow, which are based on this assumption (e.g., the conclusion that the relative importance of NAO decreases with increasing return period in all locations).

  As justification for our choice of only including the regression of NAO on our model threshold ($u$) we have updated figure 4 in the text and it is also displayed here (Fig. R1). In panels (c) and (f) we show the p-values of the quantile regression and generalized linear model to include variations of the NAO on $u$ and $\sigma$ respectively. It is shown that the quantile regression of NAO on $u$ (Fig. R1c) has a

more widespread significance with p-values widely lower than 0.05 across large parts of NW and N Europe. Conversely the generalized linear model used to show the influence of NAO on $\sigma$ is only significant in small areas with no robust indication of a significant role of the NAO. Therefore, our conclusions made that the NAO influences the location of the tail with a relatively smaller contribution on high return levels is valid for NW Europe. We have updated the text in places to comment on the significance added in Fig. 4 of the manuscript.

[Figure]

**Fig R1.** *Maps of the (a) $\beta_0$, (b) $\beta_1$, (c) $\beta_1$ p-values (d) $\alpha_0$, (e) $\alpha_1$, and (f) $\alpha_1$ p-values parameters for regressions applied to the WISC footprints from 1950–2014 with daily NAO covariate. Units are (a,d) m s$^{-1}$ and (b,e) m s$^{-1}$ st. dev$^{-1}$.*

- Please provide some additional justification/explanation in Section 3.3:
    - Please could you add some additional justification of why you use a 5-year validation period. This seems like a very short window/small sample for estimating a 200-year return period.

      The reason for the 5-year validation period is this is often the timescale of a European windstorm catastrophe model. Furthermore, reinsurance contracts are rarely longer than 5 years so this provides a limit for the timescale we would want to estimate return levels. However, in line

with comments from both reviewers we have elected to replace the analysis in section 3.3 and 3.4 with analysis featuring a 10-year validation period. This does not affect the conclusions we made in our initial manuscript, instead this choice was made in order for our results to be clearer and convey our findings more clearly. A more detailed explanation of this is found in the response below.

o   Are your results in Figure 7 conditional on the fact that you calculate the MSE based on a baseline 5-year period? Would this change if you used e.g. a 10-year window, changing your conclusions?

The conclusions are not overly sensitive to the choice of baseline period. The results presented in figure 7 can be compared to Fig. R2 below (note different y-axis scale) where we repeat the analysis with a 10-year window. We obtain the same conclusions in that the shortest catalogues have the highest MSE and this declines quickly to a more consistent value after a catalogue length of 10-15 years. There is less year-to-year variation in the MSE as catalogue length increases when using a 10-year validation period as this produces a more robust estimates of the 200-year return level due to an increase in the number of storms in the calculations. When using a 10-year validation window the MSE is slightly lower than using a 5-year window, but this is not unexpected as there is a greater amount of data used to create the 200-year return level estimates so there is likely to be less of a role of natural/inter-annual variability. The MSE values after a catalogue length of 10-15 years in Figs 7 and R2 and both are within the confidence intervals of one another. As in Fig. 7, the MSE in Fig R2 does not reduce to zero due to the unresolved natural variability that is present at all catalogue lengths. As a result of these clearer results we have elected to replace the results in sections 3.3 and 3.4 (Figs. 7 and 9) with analysis featuring a 10-year validation period. Using the simulated data, we also make conclusions consistent with our initial analysis. Figures and text have been updated throughout the revised manuscript.

[Figure]

[Figure]

[Figure]

*Fig R2.* *Mean square error of the 200-year return level estimation of different length historical catalogues against a subsequent 10 year period from the WISC catalogue for (a) Bergen, (b) London, and (c) Madrid. Solid black line shows the median mean squared error when using all possible periods. The dark and light gray areas represent the 50% and 95% confidence interval on the standard error respectively.*

o The increase in variability in the MSE in Madrid with catalogue length is not mentioned. Do you have any thoughts on this you could add? E.g., Is this due to the difference in NAO phase between the two periods being compared (e.g., similar to the results in section 3.4)?

The increased variability in MSE is most likely a sampling artefact due to fewer samples being available at longer catalogue lengths. To have a 30-year catalogue the start year must be after 1980, and with each increase in catalogue length, this start year gets later, and so the number of possible samples reduces and the confidence intervals widen.

o Would the conclusion that 'historical catalogues longer than 5-15 years do not yield improvements in the return level estimation' be made if a different validation period length were used, or is this statement relative to the 5-year based estimate? Please could you clarify? I.e., Is the conclusion of this section that only 5-15 years of footprints are needed to reliably estimate the 200-year event? I'm not sure I am convinced of this from this analysis.

As shown in Fig. R2 and R5 the conclusions made are consistent when using a longer validation length of 10 years. Therefore, the key conclusion we have made does not need to be updated. In Fig. R5 there is now a reduction in MSE at 50 years for the 200-year MSE at Bergen suggesting the sampling of a full NAO cycle is preferable. However, at 20-year and 70-year catalogue lengths the confidence intervals do overlap so we state that at least 20 years is required, but for this case a full sampling of the NAO cycle may be preferable. At London and Bergen there are no changes to the conclusions made using the earlier analysis. We have updated section 3.4 accordingly. Throughout this analysis, as our conclusions are still the same there are no major changes to our key findings. We have also updated the text to discuss the role of unresolved natural variability. A data catalogue of at least 20 years is required to have the lowest error in the estimate of the 200-year return level. This is because the MSE never reduces to zero and so even with 10 years, or 40 years of data there is still an error in the return level estimation. Therefore, there is always some natural/inter-annual variability present that is not captured in our model and we neglected to mention this in enough detail in the initial manuscript.

- Please provide more clarity in Section 3.5:

  o The exploration of a future NAO state of +1.5 is interesting, but please can you make it clearer that this is a theoretical demonstration rather than a value that has been derived from climate projections. E.g., in the conclusion (line 283) you state 'under future NAO states...', which implies over-confidence in this evolution to a +1.5 state.

    We have updated the text to be clearer that these are theoretical estimates of the future NAO state and not assumed future states. Furthermore, we stipulate that this is an upper-estimate based on the historical trend quantified by Blackport & Fyfe (2022).

    *Blackport, R. and Fyfe, J. C., Climate models fail to capture strengthening wintertime North Atlantic jet and impacts on Europe.Sci. Adv.8,eabn3112(2022).DOI:10.1126/sciadv.abn3112*

  o I am not totally clear on the approach taken in this section and which parts of Figure 10 relate to which method. There is the discussion of 'simulated events' and 'theoretical return levels'. Am I right in thinking that the same approach as in Section 3.4 is used to simulate events with varying NAO state and these make up the box plots, and then the approach in Appendix B is used to calculate the theoretical return levels shown by the red crosses? My confusion comes from lines 251-253 where you reference Appendix B when talking about the event simulation. Please could you clarify this and provide more detail in these lines.

    The reviewer is correct as to which parts of Figure 10 relate to what parts of the paper. We simulate the events as in section 3.4 but with a different definition of NAO states to account for natural variability in the NAO state. We agree that this is not clear enough and have re-written this section accordingly.

  o In the text you describe simulating events with reference NAO 0 and +1.5, but the plots in Figure 10 also include a box plot for reference NAO -1.5, this is slightly confusing. It is also a bit confusing using sigma again in this part of the method (because you use sigma to represent the tail scale parameter earlier in the manuscript). Could this be changed to another symbol?

    We agree that the use of the -1.5 NAO state is not essential or required in the presentation of the figure so have now removed this. We also agree that the use of sigma in the figure is confusing and have replaced this with the wording 'standard deviations' for clarity. Furthermore, we

have now included the Madrid data in Figure 10 and removed it from the supplement for completeness.

**OTHER MINOR POINTS**

- Abstract – The abstract jumps to describing results related to the location and scale parameters but does not mention prior to this that an EVA model is used. Could this be mentioned briefly to set the scene e.g. in line 4?

  We thank the reviewer for this comment. We have now re-written the abstract so that it briefly describes the EVA approach to provide context for the location and scale parameters.

- Line 64 – '..for a 72-hour period of each ...' typo?

  We have re-written this sentence to improve clarity. It now reads *'The footprints provide spatial estimates at 4.4km resolution of the maximum 3-second wind gust at 10-metres elevation over the 72-hour period of each downscaled storm (WISC, 2017)*

- Lines 75-77, could you include references for the 'spatial Gaussian filter' method, and the validation results?

  As the WISC data is no longer hosted by Copernicus we do not have a web link or DOI that we can link to for the documentation. We have most of the documentation offline and are hoping to come to an agreement with Copernicus regarding the hosting of data and documentation. Until that point we have now included in the *Data Availability* section of the manuscript that footprints and documentation of WISC are available at the request of the authors.

- Line 83 – do you take the NAO index for the middle day of the 72-hour footprint window, or an average over the 72 hours? Could you add this detail here?

  We take the NAO index value on the middle day of the footprint. This has now been updated in the text.

- Line 85 – My understanding is that the statistical model is applied to locations separately. Please could you add this to the beginning of Section 2.3 for clarity (i.e., 'for any given location...')?

We have updated the text to state that this is done at any given location/grid point by modifying the first sentence to *'For building the statistical model to estimate high return period windstorm gusts at any given location or gridpoint across Europe…'*.

- Lines 90-91 – please could you include clarity on why you introduce a separation between footprints with and without strong winds. Is this to justify the model being suitable for WISC footprints which only include strong winds?

  A separation is introduced because the only data available for this analysis is that from WISC. As the footprints in WISC were only made for strong/impactful cyclones we only know the rate of these strong cyclones with strong gusts. As we assume that the probability of a strong gust coming from a weak storm is small, we can ignore the rate of weak storms in our return period calculations. We have updated this section of the text to clarify how WISC only includes strong storms with high gusts and therefore this assumption is valid.

- Line 98 – I appreciate that you discuss the limitations of assuming a shape parameter of zero, but maybe this could be referenced on this line too to reassure the reader you have considered this to be an assumption. The models in Fig 2 look good though, supporting this assumption – did you find a similar good fit in other locations?

  On line 98 (original manuscript) we wrote *'By assuming the tail shape parameter is zero…'*. We have examined other locations (top 5 most populated cities in Europe; Fig R3 below) and the return level fits with zero shape parameter also appear very reasonable. In the text we have mentioned that we find similarly successful model fits at other locations.

[Figure]

*Figure R3.* *Tests of the Gumbel fit for high return periods at the 3 representative locations: (a) London, (b) Berlin, (c) Madrid, (d) Rome, and (e) Rome. The gusts for each location from 1950-2014 are plotted against their return period. Dotted black line indicates the 0.7 quantile, red line indicates the gumbel fit, black dashed line is the 10-year return period and black dash-dot line is the 200-year return period. Uncertainty estimates are calculated from the 95% confidence of the σ parameter and are shown by the grey shaded region.*

- Line 110 – for completeness it would be appreciated if the combining and rearranging of equations 1-5 to produce equation 6 could be included in the appendix.

  We have added further information in the manuscript as to how we go from equations 1-3 to get the final expression shown in equation 6.

- Figure 2 – How were the 95% confidence intervals calculated (e.g. bootstrapping)? Please could you add this detail to the caption? The same is true for a few the other figures.

  Uncertainty estimates are calculated by estimating the uncertainty of the sigma parameter. This is done by estimating quantiles (2.5, 97.5) of a gamma distribution with shape and scale parameters defined from the value of sigma and the number of footprints. We have updated the figure caption to state how the uncertainty is calculated.

- Line 175 – The manuscript notes that the 200-year return levels are higher when using the NAO covariate, and this seems to be the case for both NAO=0.5 and -0.5. Do you have a hypothesised explanation for this? Is this linked to your conclusion at the end of Section 3.5 (line 261)?

  The reason for this is due to the specified average NAO value used in the model and the quantile regression used to estimate the value of $u$. As the quantile regression results in a value of $u$ that is non-stationary with varying NAO signal there is some difference in the estimation of $u$. The average value used is 0.27 and for northern Europe this relates to a value of $u$ that is lower than the empirical $0.7^{th}$ quantile of the whole distribution (see Fig. 5). This results in a larger value of $sigma$ in the return level estimation and therefore higher 200-year return level estimates for most of Europe, particularly in northern latitudes. We have chosen not to update this in the text in order to not add too much additional information.

- I appreciated the additional plots in the supplementary material. E.g. fig S2 was useful for demonstrating the relative contributions of the components of the model for different return periods.

  Thank you we appreciate this comment

- Line 197 – '5- year validation period to quantify of..' typo?

  Thank you for spotting this typo. We have removed *'of'* in the main text.

- Line 220 – it is noted that the simulated gusts exceed the WISC dataset considerably. Could you add a statement as to whether this is to be expected, and if so, why? Is it due to the much longer simulation period?

  Yes, this is the reason. With the longer simulation the distribution is more densely populated and therefore we obtain larger values of $sigma$ than from the WISC gusts alone. We have updated the text to state that this is a result of the longer data catalogue and greater sample of the $sigma$ distribution.

- Conclusion – where appropriate please caveat the conclusions made in the bullet points based on your answers to my questions/comments above

  We have updated the conclusions throughout to update information addressed above. We specifically mention the unresolved natural variability, the fitting of our method to multiple locations across Europe, the 'theoretical' upper limit of NAO signal, and the significance of the beta parameter regression of the NAO fit on the model threshold.

- Line 304 – there as been some recent exploration of wind-gusts in the UKCP convection permitting model projection that it might be good to reference here, e.g., https://link.springer.com/article/10.1007/s00382-021-06011-4#Sec13

We chose to include references here that explicitly link increased gusts to moist processes within cyclones. As this is not explicitly studied or mentioned in the study of Manning et al. (2023) we choose not to include this reference here.

**Reviewer 2**

This paper presents a new statistical method to estimate extreme wind gusts across Europe from dataset of high resolution, historical wind storm footprints. The model allows the authors to answer interesting questions about the role of the phase of the North Atlantic Oscillation in the magnitude of extreme gust events, and the optimal length of historical dataset that is required to estimate return levels.

Please note that I am a climate scientist, not a statistician. So I leave it to other reviewers to analyse the method in critical detail, but to the best of my knowledge the method seems well formulated and is well presented. I find this study a useful addition to the literature and particuarly to the insurance community. The study is well written, the results are nicely presented, and the limitations are clearly identified. Following minor revisions below I'd be happy to accept this paper for publication.

We thank the reviewer for their kind comments. We have taken all comments on board and believe our manuscript is now considerably stronger as a result. Below we respond to each reviewer comment individually and detail the changes we have made to the manuscript.

- 'from the WISC project' could be removed from the abstract to save defining the acronym.

  We have removed this from the abstract.

- L6: 'setting the tail location parameter' and 'tail scale parameter' comes a bit out of nowhere in the abstract as we don't know what type of statistical model this is. Suggest revising how the statistical model is introduced to provide slightly more context.

  As with the comment from reviewer 1 we have revised the start of the abstract to briefly explain the extreme value model and that we are modelling the tails of windstorm gust distributions.

- L21: I'm not sure how the long-range predictability links to natural variability. Perhaps convert into multiple sentences?

  We have re-written this sentence.

- L33: You could also add (in support of this kind of framework) that if you take multiple 100 years of climate model data or century-long reanalysis to use for this exercise substantial calibration is needed, and long observational records may contain spurious trends – managing all of this is a lot of work before you even start to create a hazard event set.

  We agree with this point and thank the reviewer for these suggestions. We have expended this section and also added in the reference to Bloomfield et al. (2018).

*Bloomfield, H.C., Shaffrey, L.C., Hodges, K.I. and Vidale, P.L., 2018. A critical assessment of the long-term changes in the wintertime surface Arctic Oscillation and Northern Hemisphere storminess in the ERA20C reanalysis. Environmental Research Letters, 13(9), p.094004.*

- L36: It feels like your conclusion counteracts this point as you say ~20 year observation periods are long enough, maybe you could reflect on the record lengths you suggest and past studies on decadal variability in the conclusions?

  The reference of Thompson et al. (2017) specifically relates to quantifying the magnitude of unprecedented extremes from multiple realisations of the historical climate. We are aiming to quantify return levels based upon just our knowledge of the last 60 years of data. We accept there is a slight contradiction in our conclusions and based on this comment and others from both reviewers we have now re-worded our conclusions slightly to emphasise the fact that in our estimates a perfect estimate of the 200-year return level is never achieved and there is still some unresolved natural variability/event uncertainty regardless of the length of historical record.

- L51: Are the WISC gusts still openly available? The data availability statement implies an author must be contacted to access them. Can you gain permission to host them on a repository or encourage the original source (C3S?) to continue to host them. They're clearly useful for the community!

  These are not currently openly available. Discussions are ongoing with C3S regarding this and in time we hope that they will be available. For now we have stated the data and documentation is available from the authors.

- L54: The introduction currently doesn't mention the impact of climate change on the NAO, shifting some of the text from section 3.5 to this section would help motivate your final research question.

  We agree it would be good to discuss some of the climate change motivation in the introduction and have subsequently included information on future changes to the NAO in the introduction.

- L62: Was there any comparison of the ERA-int/ERA20C WISC footprints over the common period in their validation? It would be very useful to confirm that they are not substantial biases when the boundary forcing is changed as this would influence your results.

  There is only one footprint that has been produced using both ERA20C and ERA-interim forcing. This is for the storm on 25/1/1990 (Burns Day Storm). In Fig. R4 below we show these two footprints and the respective difference. It is notable that the core of the footprint over the south of the UK and northern France is stronger when derived from ERA-interim, with this being by over 5 m s$^{-1}$ at its

largest. There are also large differences across the Nordic Seas and to the west of the British Isles. As we only have information about one storm we cannot quantify if there are robust differences in footprints between the early and late parts of our data. Furthermore, no comparison was performed between ERA20C and ERA-interim footprints in the initial project, to the best of our knowledge. However, we do note in the manuscript now these differences which are likely a result of the resolution of the driving reanalysis. We state '*There are some differences in footprints generated from the two reanalyses, although robust differences in footprint intensity cannot be determined*'.

[Figure]

**Fig R4.** *Footprints of the Burns Day Storm (25/1/1990) derived from (a) ERA20C and (b) ERA-interim. Difference is shown in (c). Units are m s⁻¹.*

- L76: Did any issues arise in the validation of the WISC footprints that might be relevant here?

  WISC footprints (as with the predecessor project XWS) did have some issues representing gusts over high orography (Roberts et al., 2014), but these regions are not generally highly populated and so are not a major contributor to societal risk from windstorms. This is documented in the WISC documentation which we reference throughout the text and therefore we do not feel it is necessary to expand upon in our manuscript.

  *Roberts, J. F., Champion, A. J., Dawkins, L. C., Hodges, K. I., Shaffrey, L. C., Stephenson, D. B., Stringer, M. A., Thornton, H. E., and Youngman, B. D.: The XWS open access catalogue of extreme European windstorms from 1979 to 2012, Nat. Hazards Earth Syst. Sci., 14, 2487–2501, https://doi.org/10.5194/nhess-14-2487-2014, 2014.*

- L82: Just to confirm, the daily teleconnection index is calculated from monthly derived EOFs, or is it one pattern for the whole year? The sentence is currently unclear. Also do you not have data out to 2014?

The daily NAO index is derived from monthly EOFs that are then linearly interpolated to daily values. Therefore, there is a seasonal cycle and variations on the scale we are interested in. The NAO data does extend to the present day however the 1950-2000 period is used as the standardisation period on which the mean and standard deviation are calculated. We agree there is some confusion in the text and have updated this section to improve clarity with details as described above.

- L90: Can you confirm if you did any separation on the WISC footprints into 'strongest' events or if this implicit in using the extreme footprint dataset?

  No separation was performed as the WISC footprints were created for only strong storms and those that caused significant insured losses and therefore are of interest to the risk modelling community. Therefore, by definition the WISC footprints are a strong storm subset.

- L98: The discussion of the limitation around the shape parameter could be moved to here for clarity. As someone who's not an expert on these methods I found that helpful to know at this point.

  In this line we already discuss that gusts are exponentially distributed, thereby implying no upper limit. In order as to not repeat a lot of the discussion later on, we believe that this current sentence is suffice and have chosen not to make any changes.

- L101: Is the method of moments a standard technique? If so can a reference be provided?

  The method of moments is a standard, well-established method of estimating population parameters from a sample. The first moment is the mean of the population and this is estimated from our sample population. The sample in our case is the exceedances above our threshold $u$. We have added the reference of Bowman and Shenton (2006).

  Bowman, K.O. and Shenton, L.R. (2006). Estimation: Method of Moments. In Encyclopedia of Statistical Sciences (eds S. Kotz, C.B. Read, N. Balakrishnan, B. Vidakovic and N.L. Johnson). https://doi.org/10.1002/0471667196.ess1618.pub2

- L111: I followed the method well, but it might help at line 111 to quickly recap the key parameters that the method boils down to, as there are a lot of symbols introduced and it's very important for the reader to get the form of this equation clear in their mind.

  We agree and have included information as to what parameters make up the model at line 111.

- L132: Is there are a reason for the focus on the 10 year return period level? 200 years makes sense with the solvency 2 requirements but is this one driven by industry partner interest?

The 10-year return period was also chosen as it is a return period of interest to the insurance industry. It is a common return period used for validating catastrophe models and also a time period beyond which you are unlikely to have high quality claims data. Furthermore, it is a common return period for reinsurance to be attached to and finally, a time duration for which it is relatively easy to compare with observations such as station wind data. We have updated the text at line 132 to justify our choice of this return period.

- L140: sigma_hat is defined FROM the mean excess in line 101? Confirm symbols/description are correct.

This is correct. We have updated the text at line 140 to be specific as to what the parameters are defined from with regards to the distribution of wind gusts at each grid point.

- L140-146: Could you do a statistical test (e.g. pattern correlation) to confirm the NAO influence rather than the visual analysis. I had to look at this for a long time to confirm I agreed with you. To me subplots (a) and (c) look very similar but (b) looks more like (d) with the extension of high winds towards northern Spain? Is this because the 1 in 200 year return periods have a smaller influence from the NAO?

For confirming NAO influence we have now included in Fig. 4 the p-values of our quantile regression and generalized linear model of the NAO on the threshold and mean excess respectively. We show this above in Fig. R1. It can be seen how the significance of the relationship is much more widespread and coherent for the quantile regression on the threshold (beta parameters) than the generalized linear model using the mean excesses (alpha parameters). Therefore, we can conclusively say that the NAO has a stronger and more important role on the threshold and consequently the lower return levels. We now discuss this in section 3.2 and have updated the text throughout the reflect this.

- L167: 'Strong positive' relationship seems quite generous as there is still quite a spread of gust values with a single NAO index value. Can you include some more verification? (e.g. strength of correlation/model fit)

As discussed in the previous response (and shown in Fig. R1) we have calculated the p-values at all locations. In the text when discussing the model fit relating to the locations studied in Fig. 5 we now also quote the p-values. We note that the relationships are significant at Bergen ($3.5 \times 10^{-5}$) and London (0.09), but not at Madrid (0.77), where the NAO has a reduced influence on gusts.

- L174: 'Return levels are higher for the 200-year return period (Fig. 6b) than the 10-year (Fig. 6a)' This could be removed as the result is expected?

  We have elected to keep this sentence in the text as even though it is expected, we also quote the magnitude difference between the two return levels, which may be of interest to some readers.

- L194: Are the results the same if the 5-year test period is instead swapped for 2 or 10 years? This feels quite important for your result around length of historical catalogue that is required.

  We find similar conclusions when using a 10-year validation period. This is discussed above in response to reviewer 1 and shown in Fig. R2. High MSE is found for the shortest catalogues, which then plateaus after 5-15 years (as with the 5-year validation period). There are less year-to-year variations in MSE due to the greater amount of data in each validation period. Using a 2-year validation period is harder as we have ~2 footprints per year, but for estimating the return levels we use the top 30% of gusts. With only ~4 footprints in 2 years this results in a very small sample size and so it is not possible to perform analysis in this style, and we do not believe it would be representative with such strong year-to-year variations in footprint gusts. As a result of clearer figures and improved conveyance of our findings we have elected to replace the analysis of sections 3.3 and 3.4 with figures featuring a 10-year validation period. In section 3.4 the only difference is that there is now a reduction in MSE at 50 years for the 200-year MSE at Bergen (Fig. R5 and 9b) suggesting the sampling of a full NAO cycle is preferable. However, at 20-year and 70-year catalogue lengths the confidence intervals do overlap so we state that at least 20 years is required, but for this case a full sampling of the NAO cycle may be preferable. At London and Bergen there are no changes to the conclusions made using the earlier analysis. Throughout this analysis, as our conclusions are still the same there are no major changes to our key findings. We have updated figures and text throughout to reflect this change.

[Figure]

*Fig R5. Mean square error of the (a) 10-year and (b) 200-year return level estimation of different length historical catalogues against a subsequent 10 year period with events from a 1000-year simulation for Bergen. Solid black line shows the median MSE from all possible periods. The dark and light gray areas represent the 50% and 95% confidence interval on the standard error respectively. Vertical dashed grey lines indicate the periodicity of the NAO used in simulations.*

- L204: Mentioning in the methods on average how many footprints per year there are in the WISC dataset would help with the interpretation here.

  We agree and have updated the text at L90 to state that the rate of footprints is 2.1 per year on average.

- L206-210: 'Therefore, historical catalogues longer than 5-15 years do not yield improvements in the return level estimation at these three locations.' This is the only result in the paper that I'm struggling to see from the Figures. To me the uncertainty bands are large, and the MSE seems relatively similar throughout the whole period for Norway. It seems similar for years 10-40 for London, and similar throughout for Madrid. Can you provide more information on how you've defined a 'limited improvement'? And highlight that the data record needed seems very location dependent?

  We agree that some confusion arises from this section. We have re-written to state that there are no further reductions in the MSE instead of 'limited improvement'. Furthermore, we quote that there is overlap in the confidence intervals at 20 years and 70 years for the 200-year return level MSE for Bergen in section 3.4. Later in this section we note the large spread in uncertainty and year-to-year variability, which motivates our analysis using simulated wind gusts.

- L276: 'the largest gust speeds feature stochastic noise' – Can you clarify what you mean here and how it relates to the model?

  The sentence here is stating how the NAO sets the threshold for the gusts and the excess above the threshold is apparently random, due to no detectable influence from the NAO. We have elected to re-write this summary point to: *'The NAO is important for setting the threshold for gust speed return levels, with the excess above the threshold being stochastically generated.'*.

- In the abstract and conclusions it's mentioned that the framework can assess high return period losses without the complexities of a catastrophe model. I'd note that this method allows you to replace the event set component of a catastrophe model, but it doesn't get you to the losses (e.g. inclusion of exposure and vulnerability). A CAT model would still be required for that. So please revise where appropriate.

  We agree that this does not replace a catastrophe model as we do not have any damage or loss functions to convert return levels to a damage or impact metric.

However, as a method of examining return levels and benchmarking return levels from other event sets is where our method excels. We have updated the abstract and conclusions to state we do not quantify damage estimates of provide loss functions that would be useful in loss modelling. We have removed all instances of the word 'loss' or mention that we quantify losses. In the first section of the conclusion we now state *'This simple and transparent method is agnostic to the input dataset and operates without requiring calibration or the complexities of a full catastrophe models. However, it does not replace catastrophe models as no loss estimation or vulnerability module is included.'*.

- In the conclusions can you comment on how the NAO-based results could likely be used in a CAT modelling framework? Within a given winter the NAO index can vary wildly, even though it has general cycles of high/low. Given one of your results is that you only need a 20 year period for analysis, is there a preferred historical 20-year period based on historical NAO variability, or is literally any period fine?

  In a CAT modelling framework our method would be useful for providing a benchmark return level. With regards to a preferential 20-year period our results demonstrate that any is fine for examining the 200-year return level, but a large sample of data (preferably a full NAO cycle) is required for the lower (10-year) return levels. We have decided to modify the conclusions to state that even though any 20-year period can be used, there is always an error in the return level estimate that cannot be reduced (non-zero MSE in Figs. 7, 9) due to unresolved natural variability on these timescales. Therefore, consideration should always ben taken regardless of what data period is used to estimate return levels. We have added to the final paragraph of the conclusions *'Furthermore, seasonal forecasts of the NAO could be imposed to inform decision makers at shorter timescales. Consideration should be taken, however, due to the unresolved natural variability that is present in return level estimates regardless of the length of historical record.'*.

  Similarly, given the CAT modelers issues with thinking about impacts of climate change in their backward-looking models, this is a method that could be used to think about this without grappling with full climate model simulations.

  We agree and have updated the conclusions accordingly. We have added the sentence *'Our methodology allows for the assessment of theoretical future return levels without the need of climate change simulations.'*.

Figures:

- Figure 1: the colour scheme of (d) doesn't display well on my screen. Can you confirm that it's emphasisng the features you need it to?

  We are simply aiming to show where the two periods disagree in their average footprint gust speed. There is no intention to identify any specific features, just to

emphasise that there are differences that in some locations exceed 10%. We do not believe any changes are required to this figure.

- Figure 2 (and similar ones): The differences between line types are hard to distinguish (the three dashed ones). Consider revising the colours/styles?

  We have updated some of the dotted lines in Figs. 2 and 5 to be coloured instead of broken black lines.

- Figure 3: Matching colourmaps for (a) and (b) would make it easier to compare how much stronger the gusts are with increasing return period.

  Initially, we did plot (a) and (b) on the same colour scale, however, to capture all features of interest it was not possible with wide colourbar spacing to achieve this. Therefore, we elected to plot the figures on separate colorbars that reflect the changes of each return level. We have chosen not to change this figure.

- Figure 6: Including a difference plot to show the impact of the NAO states could be nice here/in the supplementary material.

  We agree and have added the difference figure to the supplementary material as Figure S3. We only show one panel as both return periods have the same difference. This is as the $\sigma(\log T...)$ component of the model cancels in the difference and the only difference is in $\beta_0 + \beta_1 x$, which is the same for both the 10-year and 200-year return levels. The main text has been updated accordingly to reference this figure.

- Figure 10: Can a different symbol be used for NAO state so not to conflict with previous use in the methods?

  We have removed the *sigma* in the caption and axis labels of Figure 10 due to this comment and the same comment from reviewer 1. We now just refer to NAO state and standard deviations.

---

## Author Response (AR2)

Reviewer response to manuscript nhess-2023-22 'Return levels of extreme European windstorms, their dependency on the NAO, and potential future risks' by Matthew D. K. Priestley, David B. Stephenson, Adam A. Scaife, Daniel Bannister, Christopher J. T. Allen, and David Wilkie

**Reviewer 2**

Thank you for your considered responses to my comments, and for revising the paper in line with these. I am largely happy with the revisions, but still have a couple of minor points that if addressed would help to clarify the approaches taken.
We thank the reviewer for their comments and further consideration of our manuscript. Below we detail our response to each of the reviewers points independently.

These are:
- I think the motivation/question you are trying to address in Section 3.3 could be better stated. I still find the relatively short 10 year validation period approach a bit confusing. You mentioned in your response that this is related to the length of a reinsurance contract (i.e. I assume you mean you want to understand how many years long a catalogue needs to be to minimise the MSE when estimating the 1 in 200 year return level for the next 10 years, as this is the typical length of the future contract and hence situation in the reinsurance sector), but you don't mention this as motivation in Section 3.3. I think this sounds like an interesting motivation and context and stating this in the paper would help to set the context of this choice of method. If this is indeed the aim here, please could this be added as explanation.
We agree with the reviewer and have added further justification at the start of section 3.3. We now state '*One factor that can contribute to the uncertainty in the estimation of the 200-year return level is the length of the historical catalogue. "This is especially important for re-insurers and their need to understand risk in the next 10 years, as this is a time horizon for business planning, and 10 years is also the typical maximum service life of a catastrophe model. The varying length of catalogue has a substantial impact on the average footprint (Fig. 1b-d) and when applied to the statistical model at a return period of 200 years, these differences are likely to be amplified. Therefore, the understanding of future risk is likely to differ with these different length catalogues.*'

- Related to this, it is interesting that your conclusions did change very slightly when using the 10 year validation period (5-15 to 10-15 years required to minimise the MSE). I wonder if the results vary if a 15, 20 etc. year validation period is used. I understand that this is then a trade off re. how many years are left for the training data in the cross validation, but if this sensitivity study has been carried out maybe the author could comment (no need to show the results) e.g., does the 10-15 year conclusion remain for other validation period lengths?
We have performed further tests using validation periods of 15 and 20 years (Fig. R1 and R2). As can be seen in Figs R1 and R2 the same pattern of MSE curve is present and the same conclusions can be drawn. The MSE reduces rapidly beyond a catalogue length of 1-5 years and then is more stable after 10-15 years. The values of MSE are different as with a longer validation period there will be less variability in the 200-year return level. However,

validation lengths longer that 10 years are not used in the re-insurance industry and therefore we elect to retain the original figure and discussion in the manuscript. We expand the text at the end of section 3.3 to state that *'Validation periods of different lengths (i.e. 5, 20 years) were also tested with conclusions being insensitive to this change (not shown).*

[Figure]

**Fig R1.** *Mean square error of the 200-year return level estimation of different length historical catalogues against a subsequent 15-year period from the WISC catalogue for (a) Bergen, (b) London, and (c) Madrid. Solid black line shows the median mean squared error when using all possible periods. The dark and light gray areas represent the 50% and 95% confidence interval on the standard error respectively.*

[Figure]

**Fig R2.** *As Fig. R1 but using a 20-year validation period*

- I like that you have set out a few distinct research questions in the introduction and I can see how the various sections can be related back to these questions, but I think this could be made clearer. I think the manuscript would be strengthened by explicitly referring back to these questions in each section of the paper, and then again in the conclusion to explicitly answer them one by one.

We thank the reviewer for their comment. We now refer back to the questions in each section of the paper. In the conclusions we state *'We raised several questions in section 1, and the key conclusions that answer these are as follows:'*

- Throughout the paper, where there are confidence intervals in plots, please can you add a description of how they are estimated/produced? E.g. in Fig 7 are these made up of all of the combinations of years with a given catalogue length?

We have updated the discussion of how confidence intervals are calculated in all of the relevant figures.

- In line 120 I think you are missing a conditional on u in the second Pr(Y>y|S)
We thank the reviewer for their comment. However, the expression for Pr(Y>y|S) is correct as written as it can be logically deduced from Eqns (2) and (3) and the definition p(u)=Pr(Y>u|S).

---

## Author Response (AR3)

Reviewer response to manuscript nhess-2023-22 'Return levels of extreme European windstorms, their dependency on the NAO, and potential future risks' by Matthew D. K. Priestley, David B. Stephenson, Adam A. Scaife, Daniel Bannister, Christopher J. T. Allen, and David Wilkie

**Editor Comments**

We thank the editor for their continued consideration of our manuscript. Below we detail our response to your raised point and changes made to the manuscript.

1. When you write that 20 years of historical data is (are) required to reduce errors in 200-year return level estimates, it is not clear what is the reference used for assessing this reduction. Please add this information both in the abstract and in the conclusions.

We thank the editor for this comment. In the work the reduced errors in 200-year return level are compared to the subsequent, and independent, 10-year period. We have updated the abstract and also the 3$^{rd}$ key point of the conclusions to state this.